# QUAD: Q-Gradient Uncertainty-Aware Guidance for Diffusion policies in Offline Reinforcement Learning

## Abstract

Diffusion-based offline reinforcement learning (RL) leverages Q-gradients of noisy actions to guide the denoising process. Existing approaches fall into two categories: (i) backpropagating the Q-gradient of the final denoised action through all steps, or (ii) directly estimating the Q-gradient of noisy actions. The former suffers from exploding or vanishing gradients as the number of denoising steps increases, while the latter becomes inaccurate when noisy actions deviate substantially from the dataset. In this work, we focus on addressing the limitations of the second category. We introduce QUAD, an uncertainty-aware Q-gradient guidance method. QUAD employs a Q-ensemble to estimate the uncertainty of Q-gradients and uses this uncertainty to constrain unreliable guidance during denoising. By down-weighting unreliable gradients, QUAD reduces the risk of producing suboptimal actions. Experiments on the D4RL benchmark show that QUAD outperforms state-of-the-art methods across most tasks.

## 1 Introduction

Reinforcement learning (RL) has achieved remarkable progress in sequential decision-making tasks, ranging from games (Mnih et al., 2013; Lample & Chaplot, 2017) to robotics (He et al., 2024b; Ze et al., 2025). However, the majority of these successes rely heavily on abundant online interactions. In many real-world domains, such as healthcare, autonomous driving, and industrial control, exploration is either prohibitively costly or inherently unsafe. Offline RL addresses this challenge by learning policies purely from pre-collected datasets (Fujimoto & Gu, 2021; Zhou et al., 2025), thereby eliminating the need for online exploration. However, it suffers from distribution shift (Levine et al., 2020): the learned policy may produce actions that deviate substantially from those observed in the dataset, resulting in unreliable value estimates and degraded performance. A key contributor to this issue is the limited expressiveness of conventional policy classes (e.g. Gaussian), which struggle to capture complex, multimodal action distributions in real-world datasets, worsening the mismatch between learned and behavior policies.

Diffusion models (Ho et al., 2020) have emerged as a powerful class of policies (Chi et al., 2023), capable of capturing highly complex action distributions and generating diverse actions. Diffusion-based offline RL methods typically combines two forms of guidance: behavior cloning (BC) guidance and Q-guidance (Wang et al., 2022). BC guidance steers the denoising trajectory towards dataset-like actions, thereby alleviating distributional shift, whereas Q-guidance leverages value estimates to promote higher-quality actions. Existing Q-guidance methods can be categorized into two classes. The first backpropagates Q-gradients from the final denoised action through all diffusion steps (Wang et al., 2022). While effective in principle, this approach suffers from vanishing or exploding gradients as the number of denoising steps increases, leading to unstable optimization. The second estimates Q-gradients of noisy actions directly at intermediate denoising steps (Fang et al., 2024), thus avoiding backpropagation through the entire trajectory. Although more stable, this method produces unreliable Q-gradients when noisy actions lie far from the data distribution, resulting in suboptimal guidance.

To address the limitations of the second class of methods, we propose **QUAD**, a Q-gradient uncertainty-aware guidance framework that improves the reliability of denoising guidance. Our

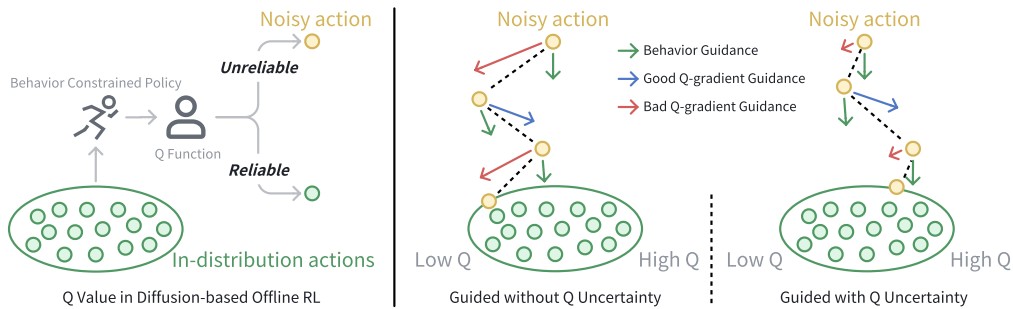

Figure 1: Left: In offline RL, behavior cloning regularization makes the learned Q-function more reliable near the dataset distribution (green), while yielding highly uncertain estimates for out-of-distribution noisy actions (orange). Right: QUAD leverages a Q-ensemble to estimate the uncertainty of Q-gradients and adaptively down-weights unreliable guidance during denoising.

key observation is that critics trained on offline data often yield highly unreliable Q estimates for noisy actions, particularly those far from the dataset distribution (Figure 1, left). To overcome this issue, QUAD employs a Q-ensemble to estimate gradient uncertainty and adaptively attenuate unreliable guidance signals (Figure 1, right). We further provide a theoretical analysis of Q-gradient uncertainty and derive an optimal weighting scheme that minimizes the alignment risk along oracle Q-gradient. Building on this analysis, we design a practical uncertainty-aware weighting mechanism that approximates the theoretical optimum. By integrating this mechanism into the Q-guidance process, QUAD effectively suppresses unreliable gradients, thereby enhancing policy performance.

We evaluate QUAD on the widely adopted D4RL benchmark (Fu et al., 2020), comparing it against state-of-the-art offline RL methods, including both non-diffusion and diffusion-based approaches. Experimental results show that QUAD consistently outperforms prior methods on most tasks and achieves comparable performance on the remaining ones.

In summary, our contributions are threefold:

- We identify and theoretically analyze the limitations of existing Q-guidance methods in diffusion-based offline RL, showing how Q-gradient uncertainty undermines reliability.
- We propose **QUAD**, a novel uncertainty-aware guidance framework that leverages a Q-ensemble to estimate gradient uncertainty and adaptively down-weight unreliable signals.
- We conduct extensive experiments on D4RL, demonstrating that QUAD achieves state-of-the-art performance across a diverse set of offline RL tasks.

## 2 PRELIMINARIES

A reinforcement learning (RL) problem is typically formulated as a Markov Decision Process (MDP), represented by the tuple $(\mathcal{S}, \mathcal{A}, \mathcal{T}, r, d_0, \gamma)$, where $\mathcal{S}$ denotes the state space, $\mathcal{A}$ the action space, $\mathcal{T}(s'|s, a)$ the transition dynamics, $r(s, a)$ the reward function, $d_0(s)$ the initial state distribution, and $\gamma \in (0, 1)$ the discount factor. The objective of RL is to learn a policy $\pi(a|s)$ that maximizes the expected discounted cumulative reward (Sutton et al., 1998):

$$J(\pi) = \mathbb{E}_{\pi, \mathcal{T}, d_0} \left[ \sum_{t=0}^{\infty} \gamma^t r(s_t, a_t) \right] \tag{1}$$

**Offline Reinforcement Learning.** Offline RL focuses on learning an effective policy solely from a fixed dataset $\mathcal{D} = \{(s_i, a_i, r_i, s'_i)\}_{i=1}^{N}$, which is generated by an (often unknown) behavior policy $\pi_\beta$, without access to further environment interactions (Levine et al., 2020). A central challenge in offline RL arises from the distributional shift between $\pi_\beta$ and the learned policy $\pi$, which can lead

to erroneous value estimates. To mitigate this issue, many approaches optimize the expected return under $Q^\pi(s,a)$ while constraining the learned policy to remain close to the behavior policy (Wu et al., 2019):

$$\max_\pi \ \mathbb{E}_{s\sim\mathcal{D},\, a\sim\pi(\cdot|s)}[Q^\pi(s,a)] \quad \text{s.t.} \quad D(\pi \,\|\, \pi_\beta) < \epsilon \tag{2}$$

where $D(\cdot,\cdot)$ denotes a divergence measure (e.g., KL divergence) and $\epsilon$ is a tolerance parameter.

**Diffusion models.** Diffusion models (Ho et al., 2020; Song et al., 2020a) are a class of generative models that assume latent variables follow a Markovian noising-denoising process. In the forward process $\{x_{0:T}\}$, Gaussian noise is gradually added to the clean data $x_0 \sim p(x_0)$ according to a predefined variance schedule $\{\beta_{1:T}\}$:

$$q(x_{1:T}|x_0) = \prod_{t=1}^{T} q(x_t|x_{t-1}), \quad q(x_t|x_{t-1}) := \mathcal{N}(x_t; \sqrt{1-\beta_t}x_{t-1}, \beta_t\mathbf{I}) \tag{3}$$

The marginal distribution admits a closed form:

$$q_t(x_t|x_0) = \mathcal{N}(x_t; \sqrt{\bar{\alpha}_t}x_0, (1-\bar{\alpha}_t)\mathbf{I}), \quad t \in \{1,\ldots,T\} \tag{4}$$

where $\alpha_t := 1 - \beta_t$ and $\bar{\alpha}_t := \prod_{s=1}^{t} \alpha_s$. Equivalently, a noisy sample can be reparameterized as

$$x_t = \sqrt{\bar{\alpha}_t}x_0 + \sqrt{1-\bar{\alpha}_t}\,\epsilon, \quad \epsilon \sim \mathcal{N}(\mathbf{0},\mathbf{I}) \tag{5}$$

Denoising diffusion probabilistic models (DDPMs) (Ho et al., 2020) parameterize the reverse process with Gaussian conditionals $p_\theta(x_{t-1}|x_t) = \mathcal{N}(x_{t-1}; \boldsymbol{\mu}_\theta(x_t,t), \boldsymbol{\Sigma}_\theta(x_t,t))$, leading to a generative process: $p_\theta(x_{0:T}) = \mathcal{N}(x_T; \mathbf{0},\mathbf{I}) \prod_{t=1}^{T} p_\theta(x_{t-1}|x_t)$. In practice, DDPMs predict the noise $\epsilon$ in Equation (5) using a neural network $\epsilon_\theta(x_t, t)$ to minimize the evidence lower bound loss:

$$\mathcal{L}(\theta) = \mathbb{E}_{x_0\sim p(x_0),\, t\sim\mathcal{U}(1,T),\, \epsilon\sim\mathcal{N}(\mathbf{0},\mathbf{I})} \left[ \|\epsilon - \epsilon_\theta(x_t,t)\|^2 \right] \tag{6}$$

**Diffusion-based Offline RL.** Following the DDPM framework, diffusion policies model action generation as a state-conditioned denoising process. Specifically, the noise predictor in (Equation (6)) is replaced with a state-conditional network $\epsilon_\theta(a^t, s, t)$ that predicts actions $a^0 \in \mathcal{A}$ given the state $s$, where $a^t$ denotes the noisy action at denoising step $t$. This formulation recovers standard behavior cloning (BC) when trained on the dataset $\mathcal{D}$. In diffusion-based offline RL, however, pure behavior cloning may fail to exploit Q value information. To address this, Q-function guidance can be incorporated to bias the denoising process toward high-value actions. A straightforward approach, as in Diffusion Q-learning (DQL) (Wang et al., 2022), backpropagates the Q-gradient from the final denoised action $a^0$ through all denoising steps, leading to the following objective:

$$\arg\min_{\pi_\theta} \mathcal{L}(\theta) = \mathbb{E}_{(s,a)\sim\mathcal{D},\, t\sim\mathcal{U}(1,T),\, \epsilon\sim\mathcal{N}(\mathbf{0},\boldsymbol{I})} \left[ \|\epsilon - \epsilon_\theta(a^t, s, t)\|^2 \right]$$
$$- \eta \cdot \mathbb{E}_{s\sim\mathcal{D},\, a^0\sim\pi_\theta} \left[ Q_\phi(s, a^0) \right] \tag{7}$$

where the first term corresponds to the denoising objective, and the second term encourages the policy to generate actions with high Q-values. The coefficient $\eta$ is a hyperparameter that balances behavior cloning against Q-guidance. An alternative strategy, as in DAC (Fang et al., 2024), directly estimates the Q-gradient of noisy actions at each denoising step, leading to the following objective:

$$\arg\min_{\pi_\theta} \mathcal{L}(\theta) = \mathbb{E}_{(s,a)\sim\mathcal{D},\, t\sim\mathcal{U}(1,T),\, \epsilon\sim\mathcal{N}(\mathbf{0},\boldsymbol{I})} \Big[ \eta \cdot \|\epsilon - \epsilon_\theta(a^t, s, t)\|^2$$
$$+ w(t) \cdot \epsilon_\theta(a^t, s, t) \cdot \nabla_{a^t} Q_\phi(s, a^t) \Big] \tag{8}$$

where $w(t)$ is a step-dependent weight that controls the influence of Q-gradient guidance across denoising steps. Rather than propagating gradients across the full sequence of denoising steps, DAC-style methods reduce the risk of vanishing or exploding gradients, thereby providing more stable optimization.

## 3 METHODS

We now introduce our proposed method, QUAD, which comprises three main components: (1) a theoretical derivation of an uncertainty-aware weighting scheme for Q-gradient guidance; (2) the formulation and implementation of a Q-gradient uncertainty-aware guidance mechanism; and (3) a practical yet principled procedure for policy extraction. An overview of the QUAD framework is shown in Figure 2.

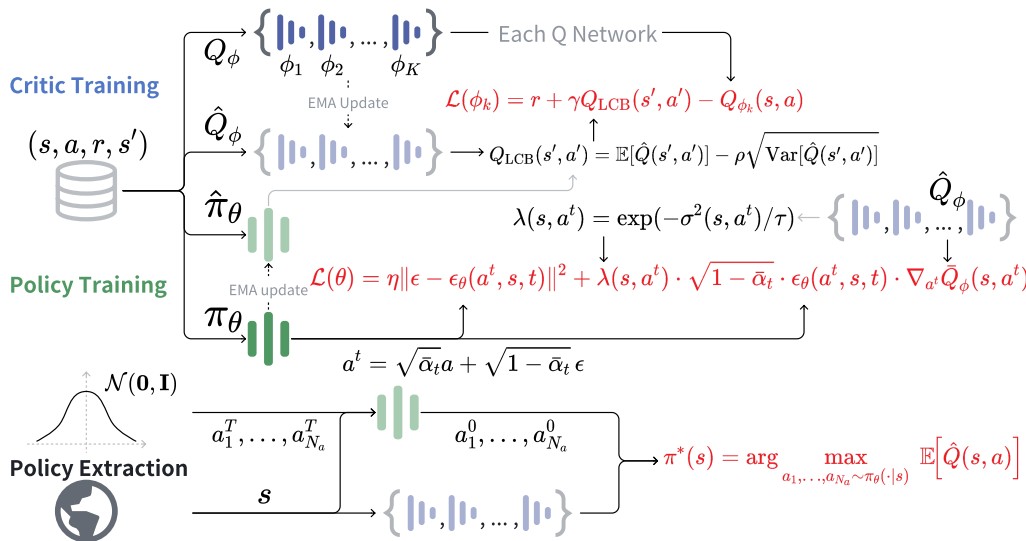

Figure 2: Overview of the QUAD framework: (1) In critic ensemble training, the target policy generates next-step actions and updates critics by minimizing TD error with LCB regularization. (2) In diffusion policy training, the target Q-ensemble estimates Q-gradients and their uncertainty to adaptively down-weight unreliable guidance. (3) In policy extraction, the target diffusion policy proposes candidate actions, and the action with the highest Q-value under the target Q-ensemble is selected.

## 3.1 THEORETICAL ANALYSIS OF Q-GRADIENT UNCERTAINTY-AWARE WEIGHTING

### 3.1.1 Q-GRADIENT UNCERTAINTY AND OPTIMAL WEIGHTING

For convenience, we denote the oracle Q-gradient as $\boldsymbol{g}^* = \nabla_{\boldsymbol{a}^t} Q^*(\boldsymbol{s}, \boldsymbol{a}^t)$. In the ideal case, the second term in Equation (8) encourages $\boldsymbol{\epsilon}_\theta$ to align with $-\boldsymbol{g}^*$, and we define the alignment loss via their inner product:

$$g^* \triangleq \boldsymbol{\epsilon}_\theta \cdot \boldsymbol{g}^*. \tag{9}$$

However, we can only access an approximation of $\boldsymbol{g}^*$, denoted as $\boldsymbol{g}_\phi = \nabla_{\boldsymbol{a}^t} Q_\phi(\boldsymbol{s}, \boldsymbol{a}^t)$. Its alignment loss along $\boldsymbol{\epsilon}_\theta$ is written as

$$g_\phi \triangleq \boldsymbol{\epsilon}_\theta \cdot \boldsymbol{g}_\phi. \tag{10}$$

We assume (see Appendix A) that $\boldsymbol{g}_\phi$ follows the biased–noisy decomposition:

$$\boldsymbol{g}_\phi = \boldsymbol{g}^* + \boldsymbol{b} + \boldsymbol{\xi}_\phi, \tag{11}$$

where $\boldsymbol{b}$ is a deterministic bias determined only by the offline dataset and the learning algorithm, and $\boldsymbol{\xi}_\phi$ is a zero-mean random noise with finite covariance, arising from stochastic function approximation and training randomness rather than from the fixed data or algorithm design. Their alignment losses along $\boldsymbol{\epsilon}_\theta$ are

$$b \triangleq \boldsymbol{\epsilon}_\theta \cdot \boldsymbol{b}, \qquad \xi_\phi \triangleq \boldsymbol{\epsilon}_\theta \cdot \boldsymbol{\xi}_\phi,$$

so that

$$g_\phi = g^* + b + \xi_\phi. \tag{12}$$

The combined term $b + \xi_\phi$ captures the epistemic uncertainty of the Q-gradient induced by limited offline data and critic approximation, which we refer to as the **Q-gradient uncertainty**. Let $\sigma^2(\boldsymbol{s}, \boldsymbol{a}^t, t) = \text{Var}(\xi_\phi)$ denote the variance of this random alignment noise at $(\boldsymbol{s}, \boldsymbol{a}^t, t)$.

Although we cannot directly reduce the Q-gradient uncertainty, we can mitigate the risk of using $g_\phi$ in the alignment loss. We define the alignment risk as the expected squared error between $g_\phi$ and $g^*$:

$$\mathcal{R} \triangleq \mathbb{E}\big[(g_\phi - g^*)^2\big], \tag{13}$$

where the expectation is taken over the randomness in $\xi_\phi$. To reduce this risk, we introduce a per-sample weighting factor $\lambda(\boldsymbol{s}, \boldsymbol{a}^t, t)$ and consider the risk of the weighted alignment loss:

$$\mathcal{R}(\lambda) \triangleq \mathbb{E}\big[(\lambda g_\phi - g^*)^2\big]. \tag{14}$$

Under the biased–noisy model above, the **optimal weighting** that minimizes $\mathcal{R}(\lambda)$ admits the closed-form solution (details see Appendix A)

$$\lambda^*(\boldsymbol{s}, \boldsymbol{a}^t, t) = \frac{g^*(g^* + b)}{(g^* + b)^2 + \sigma^2(\boldsymbol{s}, \boldsymbol{a}^t, t)}. \tag{15}$$

The sign structure of the oracle and biased term plays a crucial role:

- If $g^*$ and $g^* + b$ have the same sign (the approximate critic is directionally aligned), then $\lambda^* \in (0, 1)$ and behaves as a shrinkage factor.
- If they have opposite signs (directionally misaligned), then $\lambda^*$ may become negative or larger than 1, which is not reliably implementable without oracle access.

These two regimes are illustrated on a toy bandit example in Figure 3, where the wrong-direction regime exhibits large variance and strong disagreement across critics. This motivates the variance-only weighting strategy used in QUAD, described next.

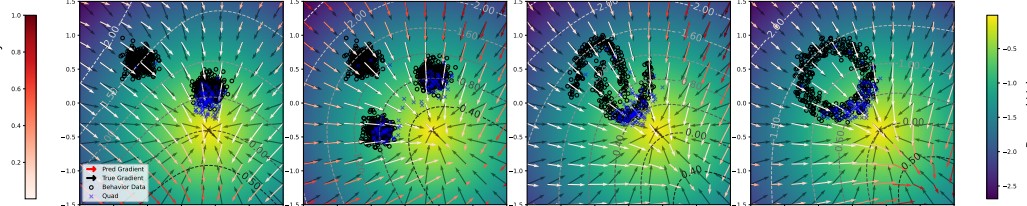

Figure 3: Bandit examples under four different data distributions: the highest reward is located at point $(0.4, -0.4)$, and QUAD (blue) can generate higher-reward actions that remain within the behavior support (black circle). Uncertainty tends to be larger when there is a significant discrepancy between the predicted Q-gradient (red or white arrow) and the ground-truth Q-gradient (black arrow).

### 3.1.2 UNCERTAINTY-AWARE WEIGHTING IN PRACTICE

Although $\lambda^*$ provides the **optimal weighting** for the abstract **Q-gradient uncertainty** model above, it depends on the unknown quantities $g^*$ and $b$, which are not identifiable from data. In practice, we rely on a Q-ensemble to obtain multiple Q-gradient estimates and use their variance to approximate the Q-gradient uncertainty.

Concretely, the variance of the projected gradients along $\epsilon_\theta$ can be estimated from the ensemble as

$$v^2(\boldsymbol{s}, \boldsymbol{a}^t, t) = \mathrm{Var}\Big(\epsilon_\theta(\boldsymbol{s}, \boldsymbol{a}^t, t) \cdot \boldsymbol{g}_{\phi_k}(\boldsymbol{s}, \boldsymbol{a}^t)\Big), \tag{16}$$

where the variance is taken over the ensemble index $k$, and $v^2$ serves as a practical estimate of the Q-gradient uncertainty at $(\boldsymbol{s}, \boldsymbol{a}^t, t)$. Empirically (see Figure 3), $v^2(\boldsymbol{s}, \boldsymbol{a}^t, t)$ is small to moderate when the learned Q-gradients are directionally aligned with the oracle gradient, and becomes large when different critics disagree strongly or the mean direction is misaligned.

These observations support a *variance-only* weighting principle: we discard the unidentifiable term and let the per-sample weight be a monotone decreasing function of $v^2(\boldsymbol{s}, \boldsymbol{a}^t, t)$,

$$\lambda(\boldsymbol{s}, \boldsymbol{a}^t, t) = f\big(v^2(\boldsymbol{s}, \boldsymbol{a}^t, t)\big), \tag{17}$$

where $f : [0, \infty) \to (0, 1]$. In QUAD we instantiate $f$ as an exponential:

$$\lambda(\boldsymbol{s}, \boldsymbol{a}^t, t) = \exp\left(-\frac{v^2(\boldsymbol{s}, \boldsymbol{a}^t, t)}{\tau}\right), \qquad \tau > 0. \tag{18}$$

This choice has several benefits: (i) it guarantees $\lambda \in (0, 1]$ and thus avoids negative or overly large weights in the misaligned regime; (ii) for small variance, a first-order expansion gives $\lambda \approx 1 - v^2/\tau$, matching the desired shrinkage behavior of the oracle solution in the aligned regime; (iii) when the ensemble variance is large, as in the wrong-direction regime, Equation (18) drives $\lambda$ close to zero and effectively turns off misleading critic guidance; and (iv) the exponential map is smooth and numerically stable, making it easy to tune in diffusion policy training.

## 3.2 DIFFUSION POLICY WITH Q-GRADIENT UNCERTAINTY-AWARE GUIDANCE

Building on the above analysis, we implement QUAD by integrating the uncertainty-aware weighting $\lambda(\boldsymbol{s}, \boldsymbol{a}^t, t)$ into the diffusion policy training objective Equation (8), following the DAC framework:

$$
\begin{aligned}
\mathcal{L}(\theta) = \mathbb{E}_{(\boldsymbol{s}, \boldsymbol{a}^*) \sim \mathcal{D}, \, t \sim \mathcal{U}(1, T), \, \boldsymbol{\epsilon} \sim \mathcal{N}(\mathbf{0}, \boldsymbol{I})} \Big[ & \eta \| \boldsymbol{\epsilon} - \boldsymbol{\epsilon}_\theta(\boldsymbol{a}^t, \boldsymbol{s}, t) \|^2 \\
& + \lambda(\boldsymbol{s}, \boldsymbol{a}^t, t) \cdot w(t) \cdot \boldsymbol{\epsilon}_\theta(\boldsymbol{a}^t, \boldsymbol{s}, t) \cdot \nabla_{\boldsymbol{a}^t} \bar{Q}_\phi(\boldsymbol{s}, \boldsymbol{a}^t) \Big],
\end{aligned}
\tag{19}
$$

where $w(t) = \sqrt{1 - \bar{\alpha}_t}$ modulates the strength of Q-guidance according to the noise level, ensuring that the denoised action remains close to the behavior policy in later diffusion steps.

A central component of QUAD is the Q-ensemble, which provides diverse and informative gradient estimates and enables approximation of epistemic uncertainty in the biased–noisy model. Concretely, we maintain $K$ parameterized Q-networks $\{Q_{\phi_k}\}_{k=1}^K$ with corresponding target networks $\{\hat{Q}_{\phi_k}\}_{k=1}^K$. To reduce overestimation and better capture epistemic uncertainty, we train this ensemble using a pessimistic Q-learning scheme (Ghasemipour et al., 2022). We adopt a lower confidence bound (LCB) as the target value and optimize each critic with the loss

$$
\begin{aligned}
\mathcal{L}(\phi_i) = & \mathbb{E}_{(\boldsymbol{s}, \boldsymbol{a}, \boldsymbol{r}, \boldsymbol{s}') \sim \mathcal{D}, \, \boldsymbol{a}' \sim \pi_\theta} \big[ \boldsymbol{r} + \gamma Q_{\text{LCB}}(\boldsymbol{s}', \boldsymbol{a}') - Q_{\phi_i}(\boldsymbol{s}, \boldsymbol{a}) \big]^2, \\
& Q_{\text{LCB}}(\boldsymbol{s}', \boldsymbol{a}') = \mathbb{E}[\hat{Q}(\boldsymbol{s}', \boldsymbol{a}')] - \rho \sqrt{\text{Var}[\hat{Q}(\boldsymbol{s}', \boldsymbol{a}')]},
\end{aligned}
\tag{20}
$$

where $\rho \geq 0$ controls the degree of pessimism, and $\mathbb{E}[\hat{Q}]$ and $\text{Var}[\hat{Q}]$ denote the empirical mean and variance across the target critics.

Subsequently, given a pair $(\boldsymbol{s}, \boldsymbol{a})$ sampled from $\mathcal{D}$, we add noise to the action following Equation (5) to obtain $(\boldsymbol{s}, \boldsymbol{a}^t)$. We then compute the ensemble alignment loss $\{g_{\phi_k} = \boldsymbol{\epsilon}_\theta \cdot \nabla_{\boldsymbol{a}^t} Q_{\phi_k}\}_{k=1}^K$. Next, we estimate the heteroscedastic Q-gradient uncertainty at $(\boldsymbol{s}, \boldsymbol{a}^t)$ via the empirical variance of this ensemble alignment loss:

$$
v_\phi^2 = \frac{1}{K} \sum_{k=1}^K \big( g_{\phi_k} - \bar{g}_\phi \big)^2, \qquad \bar{g}_\phi = \frac{1}{K} \sum_{k=1}^K g_{\phi_k}.
\tag{21}
$$

This variance $v_\phi^2$ approximates the theoretical quantity $v^2$ in Equation (16) under the biased–noisy model and serves as a data-driven estimate of the critic's epistemic uncertainty along the policy update direction. Finally, we plug $v_\phi^2$ into the exponential variance-only rule Equation (18) to obtain the per-sample weight $\lambda(\boldsymbol{s}, \boldsymbol{a}^t, t)$, which scales the Q-gradient guidance term in Equation (19). The complete QUAD training procedure is summarized in Algorithm 1.

## 3.3 POLICY EXTRACTION

We denote $\pi_\theta(\boldsymbol{a}|\boldsymbol{s})$ as the diffusion policy trained via the denoising process with noise predictor $\boldsymbol{\epsilon}_\theta(\boldsymbol{a}^t, \boldsymbol{s}, t)$. While $\pi_\theta$ can directly generate actions, we further aim to reduce uncertainty during evaluation. To this end, we draw a small batch of $N_a$ candidate actions from $\pi_\theta(\cdot|\boldsymbol{s})$ and select the one with the highest ensemble-mean Q-value:

$$
\pi^*(\boldsymbol{s}) = \arg \max_{\boldsymbol{a}_1, \ldots, \boldsymbol{a}_{N_a} \sim \pi_\theta(\cdot|\boldsymbol{s})} \mathbb{E}\Big[ \hat{Q}(\boldsymbol{s}, \boldsymbol{a}) \Big].
\tag{22}
$$

This extraction strategy is commonly employed in settings where a stochastic actor is used for critic learning, but a deterministic policy is deployed at evaluation. Because $\pi_\theta$ is already trained to approximate the target policy, only a small number of samples $N_a$ is needed. In our experiments, QUAD achieves strong performance with $N_a = 10$ following DAC, whereas SfBC and Diffusion Q-learning typically require $N_a = 32$ and $N_a = 50$, respectively.

# 4 RELATED WORK

## 4.1 OFFLINE RL

Offline RL aims to learn policies from fixed datasets, but suffers from distribution shift that leads to value overestimation in the bootstrapping process (Levine et al., 2020). To address this issue, prior works have developed strategies such as behavior regularization, conservative value estimation, and explicit Bellman error modeling. Behavior-regularized methods constrain policies to stay close to the behavior distribution via candidate-action generators or divergence penalties (Fujimoto et al., 2019; Kumar et al., 2019; Wu et al., 2019). Conservative methods alleviate OOD effects by either adding regularizers to the Q-learning objective (Kumar et al., 2020) or by learning in-sample conservative value functions (Kostrikov et al., 2021; Xu et al., 2023). Alternative approaches explicitly model Bellman errors with a Gumbel distribution and directly learn soft value functions without requiring action sampling (Garg et al., 2023). Our work is related to both behavior-regularized and conservative approaches, as we model the behavior distribution with a diffusion policy and mitigate overestimation bias via a pessimistic Q-ensemble.

## 4.2 DIFFUSION MODELS

Diffusion models are a class of generative models that consist of a forward diffusion process and a reverse denoising process (Ho et al., 2020), which can also be interpreted as stochastic differential equations (Song et al., 2020b). In the forward process, Gaussian noise is gradually added to the data according to a variance schedule. In the reverse process, a neural network is trained to predict the noise and iteratively recover the clean data. Several works improve efficiency by reducing the number of denoising steps (Song et al., 2020a; Nichol & Dhariwal, 2021; Song et al., 2023). Others explore alternative guidance strategies, such as classifier guidance (Dhariwal & Nichol, 2021) and classifier-free guidance (Ho & Salimans, 2022). More recently, diffusion models have been extended to sequential decision-making, where they are used to represent policies or trajectories (Janner et al., 2022; Chi et al., 2023; Black et al., 2023). Our work builds on diffusion policies and introduces a novel uncertainty-aware Q-gradient guidance mechanism to enhance policy learning in offline RL.

## 4.3 DIFFUSION-BASED OFFLINE RL

Diffusion-based offline RL combines diffusion models with offline RL techniques. A straightforward approach performs behavior cloning with diffusion models and then applies value-based selection to choose high-value actions from the diffusion prior (Chen et al., 2022; Hansen-Estruch et al., 2023). To reduce multi-step sampling cost, an efficient variant distills the diffusion prior into a one-step Gaussian policy (Chen et al., 2024). Another line of work integrates Q-value information directly into diffusion policy training (Wang et al., 2022). However, this approach requires backpropagating Q-gradients through the entire denoising chain, which often causes vanishing or exploding gradients. A more refined strategy applies Q-gradient guidance at each intermediate denoising step, rather than through all steps, as in DAC (Fang et al., 2024). Subsequent extensions incorporate advantage modules or pathwise regularization to further stabilize training (Chen et al., 2025; Gao et al., 2025). Despite their improved stability, these methods still suffer from unreliable Q-gradients when noisy actions deviate from the dataset distribution. Our method addresses this limitation by employing a Q-ensemble to estimate gradient uncertainty and suppress unreliable guidance, thereby improving the robustness of diffusion-based offline RL.

# 5 EXPERIMENTS

In our experiments, we aim to address the following questions:

- Does QUAD outperform state-of-the-art offline RL methods across diverse tasks?

- What is the effect of uncertainty-aware weighting on policy learning and performance?

- How sensitive is QUAD to the choice of uncertainty temperature $\tau$?

## 5.1 SETUP

**Offline Datasets.** We evaluate QUAD on the widely used D4RL benchmark (Fu et al., 2020), which covers a variety of continuous control tasks with different dataset compositions. Specifically, we consider standard locomotion tasks (HalfCheetah, Hopper, Walker2d) and the more challenging AntMaze tasks. For locomotion, we use version "v0" datasets of three quality levels: medium (m), medium-replay (m-r), and medium-expert (m-e). For AntMaze, we use version "v2" datasets: umaze (u), umaze-diverse (u-div), medium-play (m-play), medium-diverse (m-div), large-play (l-play), and large-diverse (l-div).

**Baselines.** We compare QUAD against a range of offline RL methods, including both non-diffusion and diffusion-based approaches. Non-diffusion baselines include One-step RL (Brandfonbrener et al., 2021), CQL (Kumar et al., 2020), IQL (Kostrikov et al., 2021), IVR (Xu et al., 2023) and EQL (Garg et al., 2023). Diffusion-based baselines include, Diffuser (Janner et al., 2022), SfBC (Chen et al., 2022), Diffusion Q-learning (DQL) (Wang et al., 2022), DTQL (Chen et al., 2024), AlignIQL (He et al., 2024a), and DAC (Fang et al., 2024).

**Implementation Details.** We implement QUAD on top of the publicly available DAC code-base (Fang et al., 2024). For fair comparison, we adopt the same network architectures and hyperparameters as DAC for both the diffusion policy and the Q-ensemble, unless otherwise specified. We set the ensemble size to $K = 10$ and the temperature to $\tau = 1.0$ for uncertainty weighting, based on preliminary tuning. All models are trained for 2 million gradient steps and evaluated every 20,000 steps using 10 episodes per evaluation. We report the average normalized scores over 4 random seeds for each task, and the final results are averaged over the last 5 evaluations, which typically exhibit stable performance, following the DAC protocol. For baselines, we use the results reported in their respective papers. A complete summary of experimental configurations is provided in Appendix B.

## 5.2 MAIN RESULTS

As shown in Table 1, QUAD outperforms most baselines across a variety of tasks, demonstrating the effectiveness of uncertainty-aware weighting and Q-ensemble learning. We also observe that QUAD does not achieve the best performance on AntMaze "large" tasks, where the challenges of sparse rewards and limited optimal trajectories persist. We hypothesize that in these datasets, learning a reliable critic is particularly difficult, making uncertainty-aware weighting unstable and revealing an important limitation of QUAD.

Table 1: **Average normalized scores of QUAD vs. baselines.** Abbreviations: "m" = medium, "r" = replay, "e" = expert, "u" = umaze, "div" = diverse, "l" = large. Bold numbers denote the best scores, or the second-best if achieved by our method.

| Dataset | Onestep-RL | CQL | IQL | IVR | EQL | Diffuser | DTQL | AlignIQL | SfBC | DQL | DAC | QUAD (ours) |
|---|---|---|---|---|---|---|---|---|---|---|---|---|
| halfcheetah-m | 48.4 | 44.0 | 47.4 | 48.3 | 48.3 | 44.2 | 57.9 | 46.0 | 45.9 | 51.1 | 59.1 | **59.2** ± 0.2 |
| hopper-m | 59.6 | 58.5 | 66.3 | 75.5 | 74.2 | 58.5 | 99.6 | 56.1 | 57.1 | 90.5 | **101.2** | **100.2** ± 3.5 |
| walker2d-m | 81.8 | 72.5 | 78.3 | 84.2 | 84.2 | 79.7 | 89.4 | 78.5 | 77.9 | 87.0 | 96.8 | **100.7** ± 2.4 |
| halfcheetah-m-r | 38.1 | 45.5 | 44.2 | 44.8 | 45.2 | 42.2 | 50.9 | 41.1 | 37.1 | 47.8 | 55.0 | **55.5** ± 0.3 |
| hopper-m-r | 97.5 | 95.0 | 94.7 | 99.7 | 100.7 | 96.8 | 100.0 | 74.8 | 86.2 | 101.3 | 103.1 | **103.6** ± 0.2 |
| walker2d-m-r | 49.5 | 77.2 | 73.9 | 81.2 | 82.2 | 61.2 | 88.5 | 76.5 | 65.1 | 95.5 | 96.8 | **98.9** ± 1.0 |
| halfcheetah-m-e | 93.4 | 91.6 | 86.7 | 94.0 | 94.2 | 79.8 | 92.7 | 89.1 | 92.6 | 96.8 | 99.1 | **100.1** ± 0.4 |
| hopper-m-e | 103.3 | 105.4 | 91.5 | 111.8 | 111.2 | 107.2 | 109.3 | 107.1 | 108.6 | 111.1 | 111.7 | **111.9** ± 0.4 |
| walker2d-m-e | 113.0 | 108.8 | 109.6 | 110.2 | 112.7 | 108.4 | 110.0 | 111.9 | 109.8 | 110.1 | 113.6 | **115.5** ± 1.0 |
| locomotion total | 684.6 | 698.5 | 749.7 | 749.7 | 752.9 | 678.0 | 798.3 | 681.1 | 680.3 | 791.2 | 836.4 | **845.6** |
| antmaze-u | 64.3 | 74.0 | 87.5 | 93.2 | 93.8 | - | 94.8 | 94.8 | 92.0 | 93.4 | 99.5 | **100.0** ± 0.0 |
| antmaze-u-div | 60.7 | 84.0 | 62.2 | 74.0 | 82.0 | - | 78.8 | 82.4 | 85.3 | 66.2 | 85.0 | **85.5** ± 16.5 |
| antmaze-m-play | 0.3 | 61.2 | 71.2 | 80.2 | 76.0 | - | 79.6 | 80.5 | 81.3 | 76.6 | 85.8 | **93.0** ± 3.3 |
| antmaze-m-div | 0.0 | 53.7 | 70.0 | 79.1 | 73.6 | - | 82.2 | 85.5 | 82.0 | 78.6 | 84.0 | **89.5** ± 2.6 |
| antmaze-l-play | 0.0 | 15.8 | 39.6 | 53.2 | 46.5 | - | 52.0 | **65.2** | 59.3 | 46.4 | 50.3 | 60.0 ± 4.2 |
| antmaze-l-div | 0.0 | 14.9 | 47.5 | 52.3 | 49.0 | - | **66.4** | 54.0 | 45.5 | 56.6 | 55.3 | 50.5 ± 19.5 |
| antmaze total | 125.3 | 303.6 | 378.0 | 432.0 | 420.9 | - | 441.4 | 474.8 | 445.4 | 417.8 | 459.9 | **479.0** |

## 5.3 ABLATION STUDIES

Since our method builds on DAC, which can be viewed as an unweighted variant of QUAD, we reproduce DAC using the same codebase and training protocol as QUAD, denoted as DAC-Rep. We evaluate both methods on locomotion tasks with "medium-replay" datasets and AntMaze "medium" and "large" tasks, as these settings are highly representative of dense- and sparse-reward regimes, respectively. All hyperparameters are kept identical except for those related to uncertainty modeling. As shown in Table 2, QUAD consistently outperforms DAC-Rep across these tasks, highlighting the effectiveness of uncertainty weighting. These improvements stem from QUAD's ability to mitigate the adverse effects of unreliable Q-gradients. The gains are particularly pronounced in the challenging AntMaze tasks, where Q-gradients are especially uncertain during denoising.

Table 2: Uncertainty weight ablation on locomotion "medium-replay" datasets and AntMaze "medium"/"large" tasks, comparing QUAD with its unweighted variant DAC-Rep. QUAD achieves higher returns, especially on AntMaze where Q-gradients are highly uncertain.

| uncertainty weight | walker2d | hopper | halfcheetah | antmaze | | | |
|---|---|---|---|---|---|---|---|
| | m-r | m-r | m-r | m-p | m-d | l-p | l-d |
| w/o. (DAC-Rep) | 98.1 ± 1.5 | 103.4 ± 0.2 | 55.3 ± 0.2 | 88.5 ± 3.0 | 82.5 ± 17.7 | 41.5 ± 24.4 | 42.5 ± 11.1 |
| w. (QUAD) | **98.9 ± 1.0** | **103.6 ± 0.2** | **55.5 ± 0.3** | **93.0 ± 3.3** | **89.5 ± 2.6** | **60.0 ± 4.2** | **50.5 ± 19.5** |

## 5.4 SENSITIVITY ANALYSIS

To examine the sensitivity of QUAD to key hyperparameters, we vary the uncertainty temperature $\tau \in \{0.1, 0.5, 1.0, 10.0\}$. We present results on "walker2d-medium" and "walker2d-medium-replay" in Figure 4, while full tasks are reported in Appendix B.4. Our findings indicate that QUAD is more sensitive to $\tau$ in "medium" than in "medium-replay", likely because the former exhibits a narrower data distribution, making uncertainty estimates less reliable than in "medium-replay".

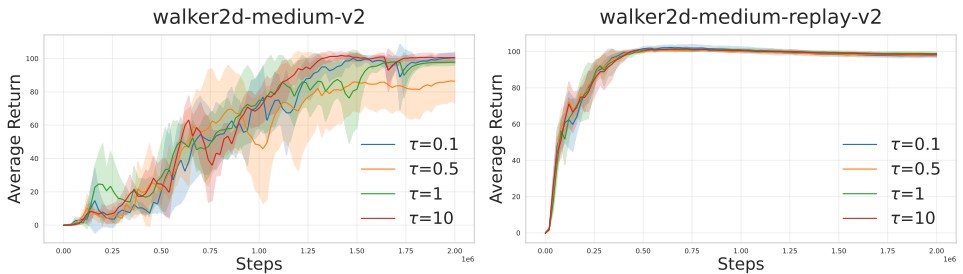

Figure 4: Sensitivity of QUAD to uncertainty temperature $\tau$, with stronger effects in "medium" due to its narrower data distribution.

## 6 CONCLUSION

We introduced QUAD, a diffusion-based offline RL method that incorporates uncertainty-aware Q-gradient weighting to improve policy learning. By leveraging a Q-ensemble to estimate uncertainty, QUAD mitigates the adverse effects of unreliable Q-gradients during denoising. Our theoretical analysis shows that this weighting scheme stabilizes optimization and enhances policy performance. Extensive experiments on the D4RL benchmark demonstrate that QUAD outperforms state-of-the-art diffusion-based methods across diverse tasks, particularly in challenging high-uncertainty settings. A limitation of QUAD lies in its reliance on the variance of Q-ensemble gradients for uncertainty estimation. The diversity of the Q-ensemble is also crucial for reliable uncertainty estimates, which may benefit from techniques such as data augmentation or ensemble diversity promotion. Future work includes exploring more advanced uncertainty estimation methods and extending QUAD to broader RL scenarios, such as offline meta-RL.

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

THE USE OF LLMS

We thank ChatGPT-5 for its assistance in polishing the writing and proofreading of this paper. The authors are responsible for the content and presentation.

## A    DETAILED PROOFS AND EXTENSIONS OF QUAD THEORY

In this appendix we complement Section 3.1 by (i) deriving the closed-form **optimal weighting** $\lambda^*$ for the biased–noisy scalar model, and (ii) explaining how the ensemble variance $v^2(s, a^t, t)$ used in practice approximates the theoretical **Q-gradient uncertainty** modeled in the main text.

### A.1    DERIVATION OF THE OPTIMAL WEIGHTING

In the main text we work with scalar alignment losses

$$g^* \triangleq \epsilon_\theta \cdot g^*, \qquad g_\phi \triangleq \epsilon_\theta \cdot g_\phi,$$

and assume the biased–noisy decomposition

$$g_\phi = g^* + b + \xi_\phi,$$

where $b$ is a deterministic bias determined only by the offline dataset and the learning algorithm, and $\xi_\phi$ is a zero-mean random noise with finite variance $\sigma^2(s, a^t, t)$, arising from stochastic function approximation and training randomness. For a fixed $(s, a^t, t)$ we drop the arguments and simply write $g^*, b, \xi, \sigma^2$.

The main text defines the alignment risk

$$\mathcal{R} \triangleq \mathbb{E}\big[(g_\phi - g^*)^2\big],$$

and, for a per-sample weight $\lambda$, the weighted risk

$$\mathcal{R}(\lambda) \triangleq \mathbb{E}\big[(\lambda g_\phi - g^*)^2\big].$$

Substituting $g_\phi = g^* + b + \xi$ gives

$$\mathcal{R}(\lambda) = \mathbb{E}\big[(\lambda(g^* + b + \xi) - g^*)^2\big]. \tag{23}$$

We now expand Equation (23) step by step. Write

$$\lambda(g^* + b + \xi) - g^* = \big(\lambda(g^* + b) - g^*\big) + \lambda\xi,$$

so that

$$\mathcal{R}(\lambda) = \mathbb{E}\big[\big(\lambda(g^* + b) - g^* + \lambda\xi\big)^2\big].$$

Using $(x + y)^2 = x^2 + 2xy + y^2$ with

$$x = \lambda(g^* + b) - g^*, \qquad y = \lambda\xi,$$

we obtain

$$\mathcal{R}(\lambda) = x^2 + 2x\,\mathbb{E}[y] + \mathbb{E}[y^2].$$

Since $\mathbb{E}[\xi] = 0$ and $\mathbb{E}[\xi^2] = \sigma^2$, we have $\mathbb{E}[y] = \lambda\mathbb{E}[\xi] = 0$ and $\mathbb{E}[y^2] = \lambda^2\sigma^2$, hence

$$\mathcal{R}(\lambda) = \big(\lambda(g^* + b) - g^*\big)^2 + \lambda^2\sigma^2. \tag{24}$$

Expanding the first term in Equation (24) yields

$$\big(\lambda(g^* + b) - g^*\big)^2 = \lambda^2(g^* + b)^2 - 2\lambda g^*(g^* + b) + (g^*)^2,$$

so

$$\begin{aligned}
\mathcal{R}(\lambda) &= \lambda^2(g^* + b)^2 - 2\lambda g^*(g^* + b) + (g^*)^2 + \lambda^2\sigma^2 \\
&= \big((g^* + b)^2 + \sigma^2\big)\lambda^2 - 2g^*(g^* + b)\lambda + (g^*)^2.
\end{aligned} \tag{25}$$

This is a quadratic function of $\lambda$ with positive leading coefficient $(g^* + b)^2 + \sigma^2 > 0$, so it is strictly convex.

Taking the derivative of Equation (25) with respect to $\lambda$ and setting it to zero:

$$\frac{\partial \mathcal{R}(\lambda)}{\partial \lambda} = 2\big((g^* + b)^2 + \sigma^2\big)\lambda - 2g^*(g^* + b) = 0,$$

we obtain the **optimal weighting**

$$\lambda^* = \frac{g^*(g^* + b)}{(g^* + b)^2 + \sigma^2}, \tag{26}$$

which is exactly the form used in the main text (with the dependence on $(\boldsymbol{s}, \boldsymbol{a}^t, t)$ made explicit there).

For completeness, evaluating Equation (25) at $\lambda = 1$ gives

$$\mathcal{R}(1) = (g^* + b)^2 + \sigma^2 - 2g^*(g^* + b) + (g^*)^2 = b^2 + \sigma^2.$$

Since Equation (25) is strictly convex and minimized at $\lambda^*$, we have $\mathcal{R}(\lambda^*) \leq \mathcal{R}(1) = b^2 + \sigma^2$, showing that there always exists a scalar weight that performs no worse (in alignment risk) than using the unweighted critic.

### A.2 WHAT DOES THE ENSEMBLE CAPTURE, AND HOW TO USE IT IN PRACTICE?

The scalar model in Section 3.1 parameterizes the **Q-gradient uncertainty** at a given $(\boldsymbol{s}, \boldsymbol{a}^t, t)$ by a deterministic bias $b$ and a noise term $\xi$ through

$$g_\phi = g^* + b + \xi,$$

where $b$ is a deterministic bias determined only by the offline dataset and the learning algorithm, and $\xi$ is zero-mean random noise with variance $\sigma^2(\boldsymbol{s}, \boldsymbol{a}^t, t) = \mathrm{Var}(\xi)$, arising from stochastic function approximation and training randomness rather than from the fixed data or algorithm design. The combined term $b + \xi$ captures the epistemic uncertainty of the Q-gradient, which we refer to as the **Q-gradient uncertainty**. In this model, the **optimal weighting** in Equation (26) depends on both $b$ and $\sigma^2(\boldsymbol{s}, \boldsymbol{a}^t, t)$.

In practice, QUAD cannot observe $b$ or $\sigma^2(\boldsymbol{s}, \boldsymbol{a}^t, t)$ directly, and instead uses a Q-ensemble to approximate the **Q-gradient uncertainty**. We maintain $K$ critics $\{Q_{\phi_k}\}_{k=1}^K$ and define their gradient-based alignment losses along $\boldsymbol{\epsilon}_\theta$ as

$$g_k(\boldsymbol{s}, \boldsymbol{a}^t, t) \triangleq \boldsymbol{\epsilon}_\theta(\boldsymbol{a}^t, \boldsymbol{s}, t) \cdot \nabla_{\boldsymbol{a}^t} Q_{\phi_k}(\boldsymbol{s}, \boldsymbol{a}^t), \qquad k = 1, \ldots, K.$$

Under the same biased–noisy picture, each $g_k$ can be written as

$$g_k = g^* + b + \xi_k,$$

where the oracle alignment $g^*$ and the deterministic bias $b$ are shared across critics at $(\boldsymbol{s}, \boldsymbol{a}^t, t)$, while the zero-mean noises $\xi_k$ come from the randomness of individual function approximators and training.

To obtain a data-driven estimate of this noise level, we use the empirical variance of the alignment losses across the ensemble:

$$v^2(\boldsymbol{s}, \boldsymbol{a}^t, t) = \frac{1}{K}\sum_{k=1}^K \big(g_k(\boldsymbol{s}, \boldsymbol{a}^t, t) - \bar{g}(\boldsymbol{s}, \boldsymbol{a}^t, t)\big)^2, \qquad \bar{g}(\boldsymbol{s}, \boldsymbol{a}^t, t) = \frac{1}{K}\sum_{k=1}^K g_k(\boldsymbol{s}, \boldsymbol{a}^t, t).$$

Under the usual assumption that the noises $\{\xi_k\}$ are independent across $k$, $v^2(\boldsymbol{s}, \boldsymbol{a}^t, t)$ is a standard sample-variance estimator of $\mathrm{Var}(g_k)$ and provides a practical approximation of the Q-gradient uncertainty that appears in the **optimal weighting** $\lambda^*$.

However, the exact form of $\lambda^*$ in Equation (26) depends on terms which are not identifiable from data. Motivated by this, QUAD adopts a variance-only weighting rule that uses $v^2(\boldsymbol{s}, \boldsymbol{a}^t, t)$ as a surrogate for the **Q-gradient uncertainty** and defines

$$\lambda(\boldsymbol{s}, \boldsymbol{a}^t, t) = f\big(v^2(\boldsymbol{s}, \boldsymbol{a}^t, t)\big),$$

where $f : [0, \infty) \to (0, 1]$ is a decreasing function. In the main text we instantiate $f$ as the exponential

$$\lambda(\boldsymbol{s}, \boldsymbol{a}^t, t) = \exp\left(-\frac{v^2(\boldsymbol{s}, \boldsymbol{a}^t, t)}{\tau}\right),$$

as in Equation (18). This choice preserves the qualitative behavior of the theoretical **optimal weighting**: when the ensemble variance (and thus the estimated **Q-gradient uncertainty**) is small, $\lambda$ stays close to 1 and preserves strong Q-gradient guidance; when the ensemble variance is large, $\lambda$ decays towards 0 and automatically down-weights unreliable critic signals.

## B  Details of Experimental Setup

We train all models for 2M gradient steps. Each environment is run with 4 independent seeds, and performance is evaluated every 20k steps using 10 additional seeds, yielding 40 rollouts per evaluation. We report the mean score over the final 50k steps without early stopping. Experiments are conducted on 4 RTX 4090 GPUs, with each run taking about 2.5 hours including training and evaluation. Our implementation builds on the jaxrl (Kostrikov, 2021) codebase.

### B.1  Network Architecture

Both the actor and critic adopt a 3-layer MLP with hidden size 256 and Mish activation (Mish, 1908). Target networks are used to stabilize training: $\hat{\epsilon}_\theta$ and $\hat{Q}_{\phi_k}$ are initialized with the same parameters as $\epsilon_\theta$ and $Q_{\phi_k}$, and track their exponential moving averages (EMA). The target actor is updated every 5 gradient steps, while the target critics are updated after each step.

### B.2  Hyperparameters

We use consistent hyperparameter settings for the diffusion models and networks across all tasks. The hyperparameters are specified as follows:

Table 3: Hyperparameters for all networks and tasks.

| Hyperparameter | Value |
|---|---|
| T (Diffusion Steps) | 5 |
| $\beta_t$ (Noise Schedule) | Variance Preserving () |
| K (Ensemble Size) | 10 |
| B (Batch Size) | 256 |
| Learning Rates (for all networks) | 3e-4, 1e-3 (antmaze-large) |
| Learning Rate Decay | Cosine () |
| Optimizer | Adam () |
| $\eta_{\text{init}}$ (Initial Behavior Cloning Strength) | [0.1, 1] |
| $\alpha_\eta$ (for Dual Gradient Ascent) | 0.001 |
| $\alpha_{\text{ema}}$ (EMA Learning Rate) | 5e-3 |
| $N_a$ (Number of sampled actions for evaluation) | 10 |
| b (Behavior Cloning Threshold) | [0.05, 1] |
| $\rho$ (Pessimistic factor) | [0, 2] |

QUAD adopts the same hyperparameters as DAC for the diffusion policy and Q-ensemble, except for AntMaze "large" tasks where a smaller $\eta$ is used. We sweep over $\tau \in \{0.1, 0.5, 1.0, 10.0, 100\}$ and report the best value for each task in Table 4.

### B.3  Pseudo code of QUAD

We provide the pseudo code of QUAD in Algorithm 1.

### B.4  Sensitivity Analysis

We present the full sensitivity analysis of QUAD to uncertainty temperature $\tau$ on all tasks in Figure 5.

Table 4: Hyperparameters settings for tasks.

| Tasks | $\tau$ | $b$ | $\eta$ | $\rho$ | Regularization Type |
|---|---|---|---|---|---|
| hopper-medium-v2 | 0.1 | 1 | - | 1.5 | Learnable |
| hopper-medium-replay-v2 | 1 | 1 | - | 1.5 | Learnable |
| hopper-medium-expert-v2 | 0.5 | 0.05 | - | 1.5 | Learnable |
| walker2d-medium-v2 | 10 | 1 | - | 1 | Learnable |
| walker2d-medium-replay-v2 | 1 | 1 | - | 1 | Learnable |
| walker2d-medium-expert-v2 | 1 | 1 | - | 1 | Learnable |
| halfcheetah-medium-v2 | 10 | 1 | - | 0 | Learnable |
| halfcheetah-medium-replay-v2 | 0.5 | 1 | - | 0 | Learnable |
| halfcheetah-medium-expert-v2 | 10 | 0.1 | - | 0 | Learnable |
| antmaze-umaze-v0 | 0.5 | - | 0.1 | 1 | Constant |
| antmaze-umaze-diverse-v0 | 0.5 | - | 0.1 | 1 | Constant |
| antmaze-medium-play-v0 | 0.1 | - | 0.1 | 1 | Constant |
| antmaze-medium-diverse-v0 | 0.1 | - | 0.1 | 1 | Constant |
| antmaze-large-play-v0 | 0.1 | - | 0.1 | 1.1 | Constant |
| antmaze-large-diverse-v0 | 0.1 | - | 0.1 | 1 | Constant |

---

**Algorithm 1** QUAD: Q-gradient Uncertainty-aware Guidance Training

---

**Require:** offline dataset $\mathcal{D}$, batch size $B$, learning rates $\alpha_\phi$, $\alpha_\theta$, $\alpha_\eta$ and $\alpha_{\text{ema}}$, behavior cloning threshold $\varepsilon_b$, pessimism factor $\rho$, initial Lagrangian multiplier $\eta_{\text{init}}$, ensemble size $K$, uncertainty temperature $\tau$

1: Initialize: diffusion policy $\epsilon_\theta$, target diffusion policy $\hat{\epsilon}_\theta = \epsilon_\theta$, Q ensemble networks $Q_{\phi_k}$, target Q ensemble networks $\hat{Q}_{\phi_k} = Q_{\phi_k}$ $(i = 1, 2, ..., K)$, Lagrangian multiplier $\eta = \eta_{\text{init}}$

2: **while** training not convergent **do**

3:     Sample a batch of $B$ transitions $\{(\boldsymbol{s}, \boldsymbol{a}, \boldsymbol{r}, \boldsymbol{s}')\} \subset \mathcal{D}$

4:     Sample $\boldsymbol{a}' = \boldsymbol{a^0}$ through denoising process using noise predictor $\hat{\epsilon}_\theta(\boldsymbol{a^t}, \boldsymbol{s}, t)$.

5:     **for** $k$ in $\{1, 2, ..., K\}$ **do**

6:         Update $\phi_k \leftarrow \phi_k - \alpha_\phi \nabla_{\phi_k} \mathcal{L}(\phi_k)$ (Equation (20))               ▷ Q ensemble learning

7:     **end for**

8:     Sample $\epsilon \sim \mathcal{N}(\mathbf{0}, \mathbf{I}), t \sim \mathcal{U}(0, T)$ and compute $\boldsymbol{a^t} = \sqrt{\bar{\alpha}_t}\boldsymbol{a} + \sqrt{1 - \bar{\alpha}_t}\epsilon$

9:     Estimate Q-gradient $\nabla_{\boldsymbol{a^t}} Q_{\pi_i}(\boldsymbol{s}, \boldsymbol{a^t})$ using (Equation (21))

10:     Estimate Q-gradient uncertainty weight $\lambda(\boldsymbol{s}, \boldsymbol{a^t})$ using (Equation (18))

11:     $\theta \leftarrow \theta - \alpha_\theta \nabla_\theta \mathcal{L}(\theta)$ (Equation (19))               ▷ Policy learning

12:     $\eta \leftarrow \eta + \alpha_\eta (||\epsilon_\theta(\boldsymbol{a^t}, \boldsymbol{s}, t) - \epsilon||^2 - \varepsilon_b)$               ▷ Dual gradient ascent (optional)

13:     $\hat{\theta} \leftarrow (1 - \alpha_{\text{ema}})\hat{\theta} + \alpha_{\text{ema}}\theta$

14:     $\hat{\phi}_i \leftarrow (1 - \alpha_{\text{ema}})\hat{\phi}_i + \alpha_{\text{ema}}\phi_i$               ▷ Update target networks using EMA

15: **end while**

---

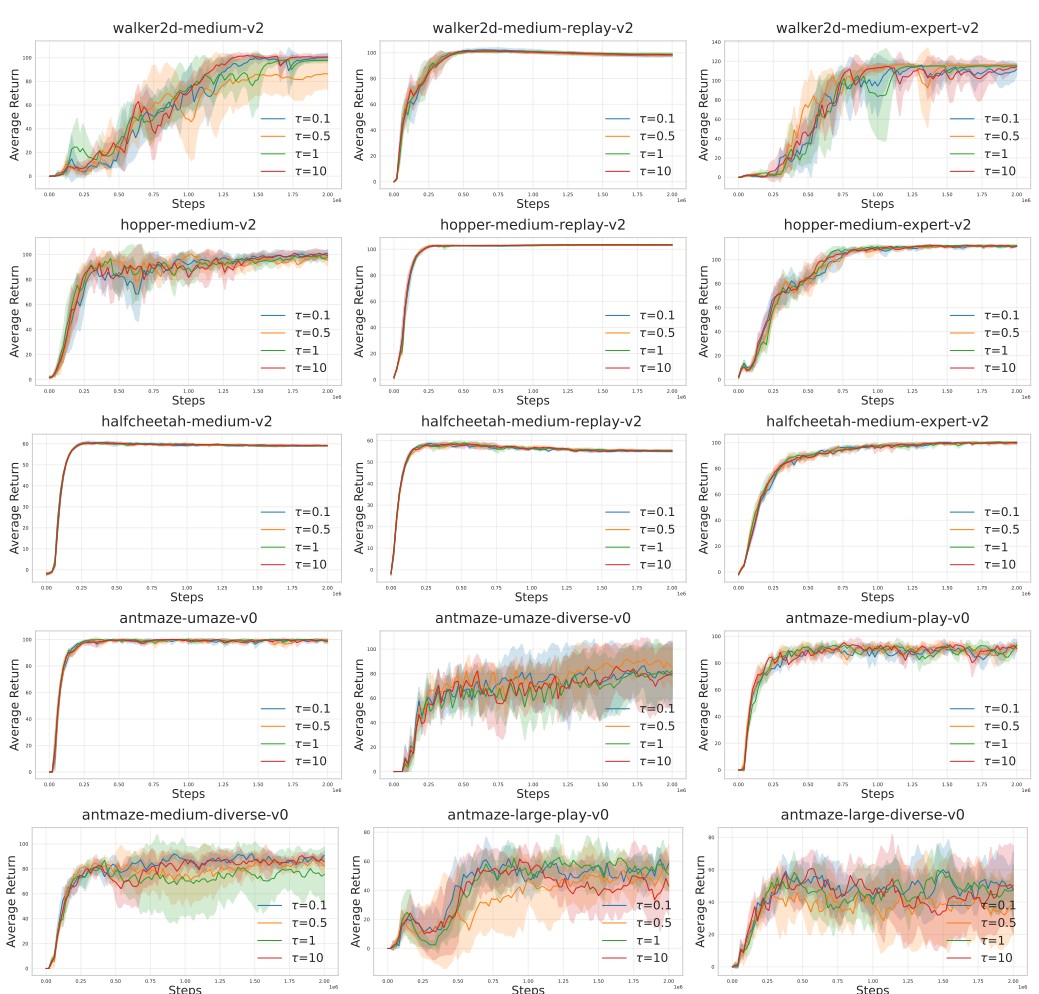

Figure 5: Sensitivity of QUAD to uncertainty temperature $\tau$ on various tasks.

