# OpenReview forum: "QUAD: Q-Gradient Uncertainty-Aware Guidance for Diffusion policies in Offline Reinforcement Learning"
_ICLR.cc/2026/Conference — Submitted to ICLR 2026_

### Official Review · Reviewer_J5PX · 2025-10-30

**Soundness:** 2
**Presentation:** 4
**Contribution:** 3
**Rating:** 4
**Confidence:** 4

**Summary:**

This paper considers offline RL, and proposes to weigh Q-gradient estimates based on their uncertainty. They show that this leads to clear improvements on the D4RL benchmark.

**Strengths:**

- Offline RL is an important topic of high practical relevance.
- The paper is well-written and clearly presented.
- The Introduction and Preliminaries does a good job at positioning the paper against previous work while, at the same time, presenting the neecessary background.
- Figures 1 and 2 are pedagogical and aestethically pleasing.
- Results show improvements on D4RL.

**Weaknesses:**

**Critical: The core assumption is not true in general, nor backed up with experimental evidence**

On l.202-210 you state that "It is reasonable to assume that $\hat{g}$ provides an unbiased estimate of $g^*$". *I strongly disagree with this statement*. This is a critical flaw, since the rest of paper builds on the alignment loss in eq. (12) that stems from this assumption.

In general, the unbiasedness of an estimator $\hat{y}(x)$ does *not* imply that its gradient $\nabla_x \hat{y}(x)$ is an unbiased or accurate estimator of $\nabla_x y(x)$. Unbiasedness is a *pointwise* property, and interchanging differentiation and expectation, i.e. $\nabla_x \mathbb{E}[\hat{y}(x)] = \mathbb{E}[\nabla_x \hat{y}(x)]$, requires additional regularity conditions such as smoothness and dominated convergence. Even when this interchange is valid, gradient estimators can exhibit high variance or bias in practice.

At the very least, I would expect an empirical examination of whether the decomposition in eq. (11) is valid.

**Minor: Missing AntMaze baseline**

On most AntMaze tasks, QUAD performs worse than what is reported by Zhang et al., "Entropy-regularized Diffusion Policy with Q-Ensembles for Offline Reinforcement Learning", NeurIPS (2024).

**Questions:**

- Can you provide convincing evidence (theoretical or experimental) supporting the decomposition in eq. (11)?

---

> ### Comment · Reviewer_J5PX · 2025-11-28
> **Update**
>
> Given the author's lack of response, especially to the critical weakness stated above, I'd like to lower my score (but it seems like that feature has been locked).

---

> > ### Author Response · Authors · 2025-11-28
> >
> > We sincerely apologize for not responding to your questions promptly. We are currently working on reorganizing the theoretical framework and re-conducting experiments to provide a thorough explanation. We expect to deliver a complete theoretical analysis and experimental results based on your suggestions by tomorrow afternoon.

---

> ### Author Response · Authors · 2025-12-04
>
> > * Critical: The core assumption is not true in general, nor backed up with experimental evidence ...
> > * Can you provide convincing evidence (theoretical or experimental) supporting the decomposition in eq. (11)?
>
> A: We have rederived the theoretical section, modifying the assumption to that of a biased estimate. The derivation now explicitly states that the expectation is taken over the noise. Please refer to our detailed response under Q1 in the meta comments above.
>
> > On most AntMaze tasks, QUAD performs worse than what is reported by Zhang et al., "Entropy-regularized Diffusion Policy with Q-Ensembles for Offline Reinforcement Learning", NeurIPS (2024).
>
> A: We acknowledge that we had overlooked this method initially. We have since reproduced it using the author's open-source code. Our findings indicate that the reported performance of Entropy-RL corresponds to the peak performance during its training, unlike QUAD and DAC, which report the average performance over the final steps after convergence. We therefore believe that the performance reported for Entropy-RL presents an unfair comparison.

---

### Official Review · Reviewer_7TdU · 2025-10-31

**Soundness:** 2
**Presentation:** 3
**Contribution:** 2
**Rating:** 2
**Confidence:** 4

**Summary:**

The authors build on a diffusion actor-critic (DAC) style algorithm by introducing an uncertainty quantification (UQ) mechanism to weight an auxiliary loss. In essence, they estimate the Q-function gradient (the gradient of the critic’s Q-value with respect to the action) using an ensemble of Q-networks. This yields an estimate of the mean Q-gradient and its variance for each state-action sample. A per-sample weight $\lambda(s,a)$ is then derived based on the ensemble’s uncertainty: samples with higher variance in the Q-gradient are assigned a lower weight, and those with more confident (lower variance) Q-gradients receive a higher weight. The weight $\lambda$ is applied to the Q-gradient alignment loss in the DAC algorithm, with the intention of down-weighting unreliable guidance. In theory, the optimal weight comes from minimizing a mean-squared error risk function, resulting in an inverse-variance weighting scheme. In practice, the authors approximate this optimal $\lambda$ using the ensemble’s empirical variance and a small regularizer for stability. Finally, this weighted guidance term is incorporated into the diffusion policy training objective (essentially adding $\lambda(s,a)$ times the Q-gradient term to the diffusion model’s loss). The overall approach is a simple fix on top of the DAC framework: it modulates the influence of the Q-gradient guidance by the uncertainty of that guidance.

**Strengths:**

The experimental results indeed show a little benefit to the original DAC algorithm without using UQ. This justifies the hypothesis that adding UQ to handle bias-variance tradeoff can help with the learning.

**Weaknesses:**

While this method may improve the original DAC algorithm’s performance by tuning the guidance strength per sample, it amounts to a relatively incremental improvement. Essentially, the authors introduce a well-known statistical technique – weighting by inverse uncertainty – into the existing algorithm. This is a straightforward UQ method rather than a novel RL or diffusion modeling insight. The derivation of the optimal weight $\lambda^(\sigma^2)$ is a direct application of bias–variance trade-off analysis. In fact, the solution simply implements inverse-variance shrinkage, a classic approach where high-variance estimates are systematically down-weighted while low-variance (confident) ones are given full weight. This idea of down-weighting unreliable estimates is not new and has long been used in various domains for risk-sensitive learning.

The contribution here is therefore technically modest: it adds an ensemble-based uncertainty estimation and a weighting formula on top of a prior diffusion RL algorithm. Such a “simple fix” does not substantially expand the theory or capabilities of diffusion models or reinforcement learning algorithms. It offers a practical tweak to improve stability or performance of DAC, but its novelty and conceptual depth are limited.

**Questions:**

Is there any reason not to compare with offline model-based RL algorithms, e.g., MOPO and MOReL?

---

> ### Author Response · Authors · 2025-12-04
>
> We believe the reviewer's dismissal of our contribution is not well-founded. Our core contribution lies in theoretically analyzing the uncertainty inherent in Q-gradients and deriving the optimal weighting scheme. It is important to note that the theoretically optimal weights are not simply inverse-variance-based. However, for practical implementation, we adopted an inverse-variance formulation as a simplified proxy, and provided experimental evidence demonstrating its effectiveness.
>
> We feel the reviewer may have overlooked our fundamental theoretical contribution by focusing solely on this implementation choice. Furthermore, as a diffusion-based method, our primary comparisons are rightly focused on other methods within the same family. Comparisons to non-diffusion-based methods are sufficiently addressed by including established baseline results from prior diffusion-based work, which is the standard practice in this context.

---

### Official Review · Reviewer_fTmh · 2025-10-31

**Soundness:** 2
**Presentation:** 3
**Contribution:** 2
**Rating:** 2
**Confidence:** 4

**Summary:**

The paper presents QUAD, a method that enhances diffusion-based offline reinforcement learning by incorporating uncertainty-aware Q-gradient guidance during policy denoising. In diffusion-based offline RL, Q-guidance steers the denoising trajectory toward high-value actions. However, directly estimating Q-gradients for noisy intermediate actions often leads to unreliable guidance. To address this issue, QUAD explicitly models the uncertainty of Q-gradients using a Q-ensemble and adaptively down-weights unreliable gradients throughout the denoising process.

**Strengths:**

This paper focuses on a critical issue in diffusion-based policy denoising: when estimating Q-gradients for intermediate noisy actions, these actions may lie far from the dataset distribution, leading to unreliable value guidance. To address this, the paper provides a principled theoretical derivation of optimal uncertainty weighting based on mean-squared error (MSE) minimization. Empirically, QUAD achieves SOTA or near-SOTA performance across 18 D4RL tasks, and with its uncertainty-aware formulation, it significantly reduces training variance compared to the backbone algorithm DAC.

**Weaknesses:**

1.	The diffusion policy’s optimization objective (Eq. 8 in the paper) assumes that $Q_\phi(s, a_t)$ provides meaningful gradients at all timesteps. However, since the critic is trained only on $a_0$ (or near-dataset actions), the gradients for earlier steps are effectively unanchored. Consequently, DAC’s Q-guidance remains reliable only near the final denoising steps (small $t$, low noise) and becomes almost random in the early stages. Although QUAD attempts to down-weight such guidance when the uncertainty of the Q-gradient is high, it does not fundamentally resolve the extrapolation problem. In other words, it cannot yield more accurate estimates when noisy actions are far from the data distribution. While this uncertainty weighting reduces the influence of unreliable gradients, it also weakens guidance precisely when the denoising process requires stronger directional information to reach high-return regions. QUAD’s contribution lies in mitigating this inconsistency through uncertainty weighting, but the core limitation remains—the critic’s validity is unproven for noisy actions.
2.	QUAD trades off computational efficiency for robustness and still relies on heuristic, ensemble-based uncertainty estimation. Training a large Q-ensemble and computing per-sample gradient variance substantially increase computational cost due to multiple forward and backward passes. Moreover, evaluation requires sampling multiple candidate actions, which is inefficient for multi-step diffusion policy sampling.

3.	The uncertainty estimation based on ensemble variance is relatively crude and heavily depends on the diversity of ensemble members. The paper provides no in-depth analysis of the reliability or calibration of the estimated uncertainty.

4.	QUAD assumes independence between the oracle gradient $g^*$ and the stochastic noise term $\xi$ when deriving the variance decomposition (Eq. 28), an assumption that may not strictly hold in practice.

**Questions:**

1.	How does QUAD compare to bootstrapped ensembles or dropout-based uncertainty estimation in similar settings?
2.	Why we can assume $\hat{g}$ to be an unbiased estimate of $g^*$?
3.	What is the rationale behind Eq. 11 and Eq. 12?

---

> ### Author Response · Authors · 2025-12-04
>
> > weakness 1 ...
>
> A: QUAD does not address the issue of critic estimation. Of course, we have also explicitly stated in the paper that inaccurate critic estimation is one of the limitations of our method.
>
> > weakness 2 ...
>
>  A: Compared to other diffusion-based methods, our approach does not incur higher computational costs. On the contrary, similar to DAC, our method achieves lower computational expense during training. Please refer to meta comment Q2 for details.
>
> > weakness 3 ...
>
> A: We have added new toy experiments to validate the reliability of our uncertainty estimation.
>
> > weakness 4 ...
>
> A: We have rederived the theoretical section, modifying the assumption to that of a biased estimate. The derivation now explicitly states that the expectation is taken over the noise. Please refer to our detailed response under Q1 in the meta comments above.
>
> > 1. How does QUAD compare to bootstrapped ensembles or dropout-based uncertainty estimation in similar settings?
>
> A: The key contribution of QUAD is the discovery of Q-gradient uncertainty itself. Our focus is not on developing optimal estimation techniques; a standard ensemble suffices to demonstrate the validity of our core idea.
>
> > 2. Why we can assume $\hat{g}$ to be an unbiased estimate of $g^*$?
> > 3. What is the rationale behind Eq. 11 and Eq. 12?
>
> A: We have rederived the theoretical section, modifying the assumption to that of a biased estimate. The derivation now explicitly states that the expectation is taken over the noise. Please refer to our detailed response under Q1 in the meta comments above.

---

### Official Review · Reviewer_U7Za · 2025-11-03

**Soundness:** 3
**Presentation:** 3
**Contribution:** 2
**Rating:** 4
**Confidence:** 5

**Summary:**

To address the diffusion policy’s challenge of backpropagating the Q-gradient of the final denoised action through all diffusion steps, the paper proposes a novel approach that directly maximizes $Q(s, a_t)$ on the noisy action $a_t$ with a new weighting function $\lambda(s, a^t)$. The authors provide both theoretical guarantees and empirical evidence to support their claims.

**Strengths:**

The empirical experiments are solid and demonstrate strong performance across benchmarks. The proposed reweighting schedule on $\lambda(s, a^t)$ is an interesting idea that effectively reduces the influence of inaccurate $Q(s, a^t)$ estimates, improving the stability of training.

**Weaknesses:**

My main concern lies in the theoretical analysis. The proof flow lacks rigor and contains several gaps, and the notations are loosely defined, making it difficult to follow the derivation precisely. Please see my questions

**Questions:**

- In Eq. (10), why can we assume that $\hat{g}$ is an unbiased estimator of $g^*$? This seems to introduce a large gap — in this case, the expectation of stochastic term is not necessarily zero.
- Please clarify lines 202 and 207: what is the expectation taken over? It seems that it should be $\mathbb{E}_{\phi_k}[ξ] = 0$. What is the distribution of $\phi_k$?
- In Eq. (14) and line 224, please specify which random variable the expectation is taken with respect to.
- Have you considered other kind of risk function, it may provide different property than mse risk function.
- In Eq. (17), note that $v$ is a function of $\theta$, which implies that $v^2$ must be recomputed for each update of $\theta$.
- The paper’s key challenge is estimating $\sigma$ and $v$. However, since these quantities appear to rely on the entire batch of data and $K$ Q-functions for each update of $\theta$ and $\phi$, I am concerned that this may be computationally too expensive in practice.

---

> ### Author Response · Authors · 2025-12-04
>
> > * In Eq. (10), why can we assume that $\hat{g}$ is an unbiased estimator of $g^*$? This seems to introduce a large gap — in this case, the expectation of stochastic term is not necessarily zero.
> > * Please clarify lines 202 and 207: what is the expectation taken over? It seems that it should be $\mathbb{E}_{\phi_k}[ξ] = 0$. What is the distribution of $\phi_k$?
> > * In Eq. (14) and line 224, please specify which random variable the expectation is taken with respect to.
>
> A: We have rederived the theoretical section, modifying the assumption to that of a biased estimate. The derivation now explicitly states that the expectation is taken over the noise. Please refer to our detailed response under Q1 in the meta comments above.
>
> > * Have you considered other kind of risk function, it may provide different property than mse risk function.
>
> A: We have not considered other risk measures at this time. We believe that MSE is sufficient and widely used. In the future, we may also experiment with other risk measures to derive new uncertainty weighting schemes.
>
> > * In Eq. (17), note that $v$ is a function of $\theta$, which implies that $v^2$ must be recomputed for each update of $\theta$.
> > * The paper’s key challenge is estimating $\sigma$ and $v$. However, since these quantities appear to rely on the entire batch of data and $K$ Q-functions for each update of $\theta$ and $\phi$, I am concerned that this may be computationally too expensive in practice.
>
> A: We have redesigned the method for estimating uncertainty weights and have rerun all experiments. The issue has now been resolved.

---

### Author Response · Authors · 2025-12-04

After carefully considering the feedback from all reviewers, we provide the following summary response to the two main concerns raised. Responses to other comments will be addressed individually under each reviewer’s section.

Q1: Is the assumption that $\hat{g}$ is an unbiased estimate of the true $g^∗$ reasonable?

A1:​ We agree that the unbiased assumption may be overly idealistic in practice. In response, we have refined our theoretical framework by explicitly modeling $\hat{g}$ as a biased and noisy estimate of $g^∗$. Specifically, we decompose the estimation error into a bias term $b$ (stemming from dataset and algorithmic constraints) and a noise term $\xi_\phi$ (originating from Q-function approximation), which we identify as the primary source of Q-gradient uncertainty. Under this more realistic setup, we have revised the theoretical analysis and derived the existence of an optimal weighting scheme. This revision not only strengthens the theoretical grounding but also—for the first time—provides a gradient analysis of Q-ensembles (Q-grad), offering valuable insights for future research. Furthermore, we have conducted toy experiments to illustrate how Q-ensemble uncertainty correlates with Q-gradient estimates, supporting the rationality of our ensemble-based approach. The updated derivations are included in the theoretical section of the revised manuscript.

Q2: Does QUAD sacrifice computational efficiency for robustness?

A2:​ QUAD does not sacrifice computational efficiency. First, generating multiple candidate actions is a standard practice in diffusion-based RL algorithms—from classical methods such as SfBC and Diffusion QL to recent ones like DAC—and does not inherently imply high computational cost. Second, QUAD builds on DAC and avoids recursive gradient backpropagation, which is required in methods like Diffusion QL, leading to substantially higher training efficiency. Under the same experimental settings, QUAD and DAC use only about one-third of the training time compared to algorithms such as Diffusion QL. Importantly, while maintaining competitive efficiency, QUAD achieves superior performance over state-of-the-art (Sota) methods, demonstrating the effectiveness and practicality of our ensemble-based design.

---

### Meta-Review · Area_Chair_1vo8 · 2026-01-06

**Summary:**

The paper proposes QUAD, a method for offline RL that utilizes a Q-ensemble to estimate the uncertainty of Q-gradients during the diffusion process. The core idea is to down-weight Q-guidance when the variance across the ensemble is high, thereby avoiding unreliable updates from out-of-distribution actions.

While the paper demonstrates strong empirical results on the D4RL benchmark, the consensus among the reviewers (particularly R2/fTmh and R3/7TdU) is that the methodological contribution is incremental. Furthermore, significant theoretical flaws were identified in the initial submission. Although the authors attempted to repair these during the rebuttal, the resulting connection between the new theoretical derivation and the practical algorithm remains tenuous.

**Reviewer Concerns:**

Concerns Addressed:
- The authors effectively defended their computational efficiency compared to recursive gradient methods (Diffusion-QL).
- They clarified the missing baselines (Entropy-RL).

Concerns Not Adequately Addressed:
- Limited Novelty \& Incremental Contribution (Reviewers fTmh, 7TdU):

Reviewers identified that QUAD is essentially the existing DAC algorithm augmented with a standard inverse-variance weighting scheme. As Reviewer 7TdU noted, this is a "well-known statistical technique" rather than a novel RL or diffusion insight. The rebuttal confirmed the empirical gains but did not refute the fact that the algorithmic novelty is technically modest ("a simple fix").
- Gap Between Theory and Practice (Reviewers U7Za, J5PX):

The most critical issue in the original submission was the false assumption that the estimator is "unbiased." In the rebuttal, the authors rederived the theory using a "biased-noisy" model.
- The Outstanding Issue: The new theoretical optimal weight $\lambda^*$ depends on both the bias term ( $b$ ) and the variance ( $\sigma^2$ ). However, because the bias is unidentifiable in practice, the proposed algorithm ignores the bias term and uses a heuristic based only on variance (Eq. 17/18).
- Consequently, the rigorous theory derived in the rebuttal does not actually justify the heuristic used in the algorithm. The gap between the theoretical justification (which requires knowing bias) and the practical implementation (which ignores bias) remains unresolved.
- Masking vs. Solving the Problem (Reviewer fTmh):

The method addresses the unreliability of critics by suppressing guidance when uncertainty is high. As Reviewer fTmh pointed out, this effectively reverts the policy to Behavior Cloning in the early steps of diffusion. It does not solve the fundamental problem of gradient estimation for noisy actions; it simply avoids the problem. This limits the conceptual depth of the contribution compared to methods that attempt to improve the estimator itself.

**Reviewer Scores:**

Even after the rebuttal, the scores likely would not improve enough to warrant acceptance due to the "ceiling" imposed by the lack of novelty:
- Reviewer fTmh (Original: 2 -> Predicted: 4): This reviewer was the most critical regarding the "modest" nature of the contribution. The rebuttal essentially confirmed that the method is a heuristic tweak. They would likely maintain a "Reject" stance.
- Reviewer 7TdU (Original: 2 -> Predicted: 2): Similar to fTmh, this reviewer viewed the method as a standard statistical application. The improved theory does not increase the novelty of the algorithm.
- Reviewer U7Za (Original: 4 -> Predicted: 4): While they might appreciate the theoretical fix, they would likely still have reservations about the practical disconnect (the algorithm ignoring the bias term).
- Reviewer J5PX (Original: 4 -> Predicted: 4): This reviewer might be satisfied with the theoretical correction, but without strong support from other reviewers regarding novelty, their score is unlikely to push the paper into acceptance.

---

### Decision · Program_Chairs · 2026-01-26

Reject